# Amino Acid Composition of Amniotic Fluid during the Perinatal Period Reflects Mother’s Fat and Carbohydrate Intake

**DOI:** 10.3390/nu13072136

**Published:** 2021-06-22

**Authors:** Mitsue Sano, Haruna Nagura, Sayako Ueno, Akira Nakashima

**Affiliations:** 1Department of Nutrition, School of Human Cultures, The University of Shiga Prefecture, Hikone 522-8533, Shiga, Japan; min.nagu03@gmail.com (H.N.); poohsu_k@yahoo.co.jp (S.U.); 2Jinno Ladies Clinic-Branch Hospital “Alice”, Hikone 522-0057, Shiga, Japan; sanoo@nifty.com

**Keywords:** maternal diet, amniotic fluid, amino acids, fetal nutrition

## Abstract

Dietary content during pregnancy is important because it is necessary for the growth of the fetus. With the assumption that the nutritional status of the fetus can be monitored by measuring amino acid concentrations in the amniotic fluid, we investigated whether the habitual dietary intake of pregnant women affected the composition of the amniotic fluid and the significance of performing amniotic fluid analysis. The subjects were 34 mothers who delivered full-term babies by cesarean section. Three biological samples were collected from the mothers: blood, cord blood, and amniotic fluid. At the same time, the mothers’ prenatal nutritional intake information was also recorded. When the amino acid contents of the samples were compared with the mothers’ nutrient intake, many amino acids in the amniotic fluid were positively correlated with lipid intake, but not with protein intake. There was a negative correlation between lipid intake and carbohydrate intake, and the amino acid contents of the amniotic fluid were also negatively correlated with carbohydrate intake. The results of this study were consistent with those found in animal models, suggesting that the analysis of amniotic fluid may be a useful method to investigate the effects of habitual diet during human pregnancy on the fetus.

## 1. Introduction

Maternal nutrition during pregnancy is important for fetal growth as well as the child’s brain development and future health [1,2,3]. There have been multiple reports from developmental origins of health and disease (DOHaD) paradigm in which newborns with low birth weight (<2500 g) have an increased risk of hypertension, type 2 diabetes, and renal disease, among other health problems [4,5,6,7,8]. Low birth weight results from either preterm birth (at <37 weeks of gestation) or intrauterine growth restriction (IUGR), which is defined as a birth weight below the 10th percentile for gestational age. For instance, infants born at full term (38 weeks) are considered to have IUGR if their birth weight is <2500 g. IUGR is caused by decreased blood flow in the placenta as well as maternal undernutrition. Therefore, animal models of IUGR are created by nutritional interventions, such as feeding mothers low protein diets and reducing the amount of feed [9]. The nutritional status of the pregnant mother affects the fetus, but the nutritional status of the mother is not directly equivalent to the nutritional status of the fetus. This is because the passage of nutrients through the placenta is selective and may be either facilitated by the placenta (with or without transporters, e.g., vitamin B_1_, B_2_, B_6_, B_12_, folate, biotin, niacin, Ca, Fe), occur through diffusion (e.g., Cr, Mg), or be inefficient (e.g., TG, vitamins K and E, Cu) [10,11,12,13,14,15]. The nutritional status of the fetus is generally evaluated indirectly based on the degree of physical development according to ultrasound examination and the nutritional status of the mother. If a biological sample could be obtained from the fetus, the nutritional status of the fetus could be directly evaluated; however, this would be highly invasive for both the fetus and the mother and there would be a risk of miscarriage, so this type of sampling is not performed.

We focused on amniotic fluid, which can be collected at birth, as a means of revealing the nutritional status of the fetus. It has been reported that, at the end of pregnancy, the fetus excretes urine in the amniotic fluid and drinks amniotic fluid, which contains carbohydrates, proteins and peptides, lipids, vitamins, and minerals [12,14,16,17,18]. In addition, it has been reported that the composition of amniotic fluid can be altered by specific interventions (e.g., feed loading) focused on the mother [16,18]. If amniotic fluid can be assessed as a biological sample in order to analyze the impact of the mother’s overall diet during pregnancy on the fetus, we can provide dietary guidance that will effectively deliver the nutrients required by the fetus. This would make it possible to provide dietary guidance to reduce the risk of lifestyle-related diseases in the offspring caused by nutritional deficiencies during the fetal period. In this study, we focused on frequently consumed protein sources, which are essential for fetal growth, and investigated whether the amino acid composition in amniotic fluid changes depending on the mother’s diet.

## 2. Materials and Methods

### Study Design and Subjects

This was a basic experimental study involving humans. The experimental protocol was approved by the ethics committee of The University of Shiga Prefecture (# 476). We recruited 36 healthy mothers who delivered full-term babies by caesarean section at the Alice branch of the Jinno Ladies Clinic Hospital (Hikone City, Shiga, Japan) from February 2016 to February 2017 (Figure 1). Two of the 36 subjects were excluded because they were unable to complete the dietary survey, resulting in a final number of 34 subjects. All participating mothers provided written informed voluntary consent. None of the subjects or babies experienced complications during delivery. We collected biological samples (maternal blood, cord blood, and amniotic fluid), antenatal daily nutrient intake for the mothers, and physical data for both the mothers and neonates.

## 3. Sample Collection

Physical data (parity of the mother and gestational age, birthweight, and sex of the neonate) were obtained from the medical records after delivery. Other physical data (maternal age, body weight, and height) were obtained from the mothers by self-declaration after delivery.

Daily nutrient intake was explored using a brief, self-administered diet history questionnaire (BDHQ), where the responses reflected the average nutrient intake over the month before delivery [19,20]. A dietitian was present when the questionnaire was filled out and explained how to fill it out, what to look out for, and answered any questions. The questionnaire included a space to indicate any supplements the mothers were taking. In addition to the written survey, the dietician asked each question verbally to confirm the responses. We calculated and analyzed maternal nutrient intake to find the following values: daily energy intake; protein, fat, and carbohydrates as percent of daily energy; and energy-adjusted salt, dietary fiber, vitamin, and mineral intake [21,22].

Biological samples, comprising 6 mL umbilical vein cord blood and 10 mL amniotic fluid, were collected during the caesarean section. In addition, 6 mL of maternal blood was collected before breakfast the day after delivery. Lithium heparin-coated blood collection tubes were used to collect the blood samples (Becton Dickinson Co., Tokyo, Japan). After collection, blood glucose and ketone body (β-hydroxybutyric acid) levels were analyzed using a Precision Xceed H instrument (Abbott Japan Co. Ltd., Tokyo, Japan), and glucose levels in amniotic fluid were measured using a portable glucose analyzer (GF-501; TANITA Corp., Tokyo, Japan). Plasma samples were separated by centrifugation (3000× *g*, 15 min at 4 °C). Amniotic fluid solids were removed by centrifugation (5000 rpm, 5 min at 4 °C). The plasma and amniotic fluid samples were stored at −80 °C until required.

## 4. Measurements

Abbreviations used: Hyp, hydroxyproline; P-Ser, o-phosphoserine; Tau, taurine; PEA, o-phosphoethanolamine; a-AAA, α-aminoadipic acid; Cit, citrulline; a-ABA, α-aminobutyric acid; cystathio, cystathionine; b-Ala, β-alanine; b-AiBA, β-amino-iso-butyric acid; g-ABA, γ-aminobutyric acid; EOHNH_2_, ethanolamine; NH_3_, ammonia; Hyl, hydroxylysine; Orn, ornithine; His(1-Me), 1-methylhistidine; His(3-Me), 3-methylhistidine; Car, carnosine.

We analyzed 41 free amino acid forms (Leu, Ile, Val, Thr, Trp, Phe, Met, Lys, His, Asn, Asp, Ala, Arg, Gly, Gln, Glu, Pro, Hyp, Ser, Tyr, Cys, P-Ser, Tau, PEA, urea, Sar, a-AAA, Cit, a-ABA, cystathio, b-Ala, b-AiBA, g-ABA, EOHNH_2_, NH_3_, Hyl, Orn, His(1-Me), His(3-Me), Car, and anserine) in biological samples by post-column colorimetric derivatization with the ninhydrin method using an amino acid auto-analyzer (L-8900; Hitachi High-Tech Corp., Tokyo, Japan). The 200 μL samples of amniotic fluid or plasma were diluted 2 times with 0.02 mol/L HCl, and 200 μL of 1.5 mol/L HClO_4_ was added to deproteinate the samples. The mixtures were centrifuged at 10,000 rpm for 1 min at 4 °C, and the supernatants were filtered using a Cosmonice Filter W (0.45 μm, 4 mm; Nacalai Tesque, Inc., Kyoto, Japan). Analysis was performed using 40 µL of the filtered samples.

## 5. Statistics

All data are presented as mean ± SD, and the significance level was set at 0.05. We compared maternal nutrient and food intake vs. the concentrations of amino acids in the three types of biological samples using JMP12.2 software (SAS Institute Inc., Cary, NC, USA) and BellCurve for Excel, version 3.0 (Social Survey Research Information Co., Ltd., Tokyo, Japan).

The amino acid concentrations in the three biological samples shown in Table 1 were analyzed using the Steel–Dwass test or Tukey’s honestly significant difference (HSD) test. The correlations between the amino acid concentrations in the three biological samples and maternal protein, fat, and carbohydrate intake were analyzed using Spearman’s rank correlation coefficient with missing values excluded in a pairwise manner (Table 2). Four amino acids (g-ABA, Hyl, His(1-Me) and Car) could not be measured in many samples as their concentrations were below the detection limit and were thus excluded from the analysis. The correlations between the amino acid contents in the amniotic fluid and the intake of different types of lipids were analyzed by Spearman’s rank correlation coefficient with missing values excluded in a case-wise manner (Table 3). The types of lipids analyzed included fatty acids (FAs), saturated fatty acids (SFAs), monounsaturated fatty acids (MUFAs), polyunsaturated fatty acids (PUFAs), n-3 fatty acids (n-3 FAs), n-6 fatty acids (n-6 FAs), palmitic acid (C16:0), heptadecanoic acid (C17:0), stearic acid (C18:0), oleic acid (C18:1), linoleic acid (C18:2(n-6)), and eicosatrienoic acid (C20:3(n-6)). The correlations between NH_3_ content in the three biological samples and maternal protein and protein-rich food intake were analyzed by Spearman’s rank correlation coefficient with missing values excluded in a pairwise manner (Table 4).

## 6. Results

The baseline characteristics for mothers and infants are shown in Table 5. Participants in this experiment were full-term infants (37–41 weeks) delivered by caesarian section from Japanese mothers with standard physique, including height and weight, the day after delivery. The average age of the mothers in this study (32.4 ± 4.6 years old) was very similar to the average age at the birth of a second child among Japanese women. Of the mothers who participated in this study, 91% were multipara because the selection criteria included delivery by caesarean section (85% of participant mothers had undergone a cesarean section for a previous birth). In Japan, caesarean sections are commonly planned to avoid the risks present in a vaginal delivery after a previous caesarean section. The average number of offspring in Japan in 2016 was 1.44, but the parity of the participants in this study was higher (2.2 ± 0.7) for this reason. The birth weight of almost all newborns was over the low-birth-weight reference weight of 2500 g (only one newborn was under the reference weight, weighing 2485 g). Nutritional condition at delivery was assessed based on blood concentrations of glucose and ketone bodies. The maternal blood concentration of glucose was higher than the fetal concentration in cord blood. In contrast, the ketone body concentration in maternal blood was lower than in cord blood.

Table 6 shows the daily nutrient intake per 1000 kcal according to the BDHQ. Our data mostly matched past reports for late-stage pregnancy that had used the BDHQ and the DHQ (self-administered diet-history questionnaire) [23,24,25]. The average energy intake (1565 ± 384 kcal/day) was well below that recommended for pregnant women during late pregnancy (2100~2750 kcal/day). We considered that energy intake was likely low due to underreporting [23,24,25]. The consumption levels of other nutrients were generally satisfactory.

The amino acid contents of each sample are shown in Table 1. In the amino acid analysis, 38 of 41 amino acids were detected (Pro, Hyp, and anserine were not detected). In general, concentrations of the nine essential amino acids were highest in cord blood plasma, moderate in maternal blood plasma, and lowest in amniotic fluid, with two exceptions: the concentration of Ala was higher in maternal blood and the concentration of Gly was the same in maternal blood and amniotic fluid.

A correlation chart is shown in Table 2. This chart compares the concentrations of each amino acid in the three biological samples to the maternal dietary intake divided into three major nutrient categories (proteins, fats, and carbohydrates). Compared with other samples, the amino acid concentration profile of amniotic fluid was enriched for the types of amino acids that were correlated with lipid (positively) and carbohydrate (negatively) intake. To identify the fatty acid species responsible for the correlation of lipid intake with amino acids in the amniotic fluid, we divided the data into lipid species (fat, animal fat, vegetable fat, cholesterol, total SFAs, total MUFAs, total PUFAs, total n-3 FAs, total n-6 FAs, and 36 specific types of FAs) The intake of each FA and the amino acid contents in the amniotic fluid were compared (Table 3). Table 3 shows 6 selected fatty acids (palmitic acid (C16:0), heptadecanoic acid (C17:0), stearic acid (C18:0), oleic acid (C18:1), linoleic acid (C18:2(n-6)), and eicosatrienoic acid (C20:3(n-6))) out of the 36 FAs analyzed (data not shown for the other 30 FAs) that either show a similar correlation with overall lipid intake or were FAs with high intake levels. The results showed that animal lipids, total MUFAs, C16:0, C18:0, and C18:1 had similar correlations as the overall lipid intake.

Additionally, we analyzed whether the concentrations of ammonia (NH_3_) as a metabolite from amino acids in the maternal plasma, cord blood plasma, or amniotic fluid samples were correlated with a maternal diet high in protein content and found that NH_3_ in amniotic fluid correlated significantly with chicken (correlation coefficient: 0.4817) and pork and beef intake (correlation coefficient: 0.3959; Table 4). We also analyzed the correlation between the amino acid content in amniotic fluid and food intake, and the correlation diagrams are shown for eggs and tsukemono (green leafy vegetable pickles), which showed high correlation (Figure 2). Three amino acids, Asp, Phe, and Tyr, showed positive correlations with egg intake (Figure 2a–c). Tsukemono showed negative correlations with many amino acids, especially the three with the highest correlation coefficients (Trp, Leu, and Ser) (Figure 2d–f).

## 7. Discussion

The BMI of the mothers was calculated from their self-reported height and weight during the dietary survey on the third or fourth day after delivery. Although the calculated BMI was over 25, we did not consider the subjects of this study to be overweight because it was reported that maternal postpartum weight begins to decrease after about the fourth day [26]. One of the limitations of this study is that the height and weight of the mothers were self-reported at the time of the dietary survey and not actual measurements taken at the time of the survey. The birth weights recorded in this study (male, 2.962 ± 0.280 kg; female, 2.974 ± 0.279 kg) were similar to the average birth weights in 2015 in Japan (male, 3.040 kg; female, 2.960 kg). The average rate of low-birth-weight infant births in developed countries is about 7%, while that of Japan is high, at 9.5% for 2011–2016, according to UNICEF statistical data. There is also an indication that the average birth weight of babies born to Japanese parents living in the United States is lower than those with parents from other Asian countries (Japanese, 3.093 kg; Chinese, 3.250 kg; Korean, 3.272 kg) [27]. Japanese babies may have lower birth weights not only due to genetic differences in the mothers, but also due to the influence of environmental factors, including eating habits and psychological factors, such as the desire to lose weight, on the mother’s nutrient intake.

Umbilical cord blood glucose levels were low compared with some literature values due to the effects of the caesarean delivery. It has previously been reported that the blood glucose concentrations in umbilical cord blood from cesarean-delivered newborns are lower than those from newborns delivered vaginally [28,29]. Umbilical cord blood ketone body levels were equivalent to those previously reported in the literature [30]. One of the limitations of this study is the delay of about half a day between the time when cord blood and amniotic fluid were collected, since maternal blood was collected the morning after delivery (before breakfast).

The daily nutrient intake values reported in this study were similar to the results obtained using the BDHQ and DHQ that have been reported in other studies (Table 6) [23,24,25]. Although the low energy intake has been discussed in previous studies and is considered to be a result of underreporting, we believe that the BDHQ is appropriate for use in this study because it was reported to be valid in studies comparing FA intake and FA blood levels in pregnant women [25]. However, one of the limitations of the study is the shortcomings of the BDHQ-based survey used, which is prone to underreporting.

The concentrations of amino acids in cord blood were generally higher than in maternal blood; only levels of Ala were higher in maternal blood (Table 1) [31]. We considered that blood Ala levels were maintained at higher levels than other amino acids during late pregnancy, in accordance with a previous report that the glucose-alanine cycle is increased to provide gluconeogenic substrates during late gestation [32].

The correlation between amino acid content in maternal blood, cord blood, and amniotic fluid samples and the maternal intake of major nutrient categories (i.e., proteins, fats, and carbohydrates) was investigated (Table 2). Maternal protein intake was not strongly correlated with the maternal blood concentration of any amino acid, although for two amino acids there was a negative correlation (Met and Arg). We consider the poor correlation between maternal blood amino acid contents and maternal protein intake to be due to the increased maternal and fetal nutritional requirements and the use of amino acids for protein synthesis. It has been reported that the human embryo can catabolize amino acids to produce energy [33]. It is also possible that the catabolic metabolism of amino acids is enhanced, as studies on pregnant rats have shown that tryptophan catabolism is increased in late pregnancy compared with mid-pregnancy [34]. Unexpectedly, amino acid concentrations in amniotic fluid were positively correlated with maternal lipid intake; conversely, they were negatively correlated with carbohydrate intake. We consider this inverse correlation to be due to the negative correlation (−0.9304) observed between lipid and carbohydrate intake.

The concentrations of many amino acids in the amniotic fluid were not positively correlated with maternal protein intake, nor was the level of the amino acid metabolite NH_3_ (Table 2). It was reported that blood NH_3_ concentration increases after protein intake depending on the amount of protein consumed [35]. However, it was reported that the increase in blood NH_3_ following consumption of a protein load is transient [36]. Our study employed a survey of habitual intake to record protein intake and blood collection was performed under fasting conditions. A correlation between the concentration of NH_3_ in the amniotic fluid and maternal protein intake was not anticipated. It was also reported that, during late pregnancy, the demand for proteins and amino acids increases, and thus amino acid catabolism and urea synthesis decrease [37]. Similar to what is seen for NH_3_, amino acids may also not correlate with maternal intake due to the increasing amino acid requirements during late pregnancy. Alternatively, NH_3_ concentrations in amniotic fluid were positively correlated with the intake of meat, especially chicken. This correlation was not seen in maternal or cord blood (Table 4). This suggests that the effect of the mother’s habitual diet may be difficult to detect in blood due to the influence of the nutritional requirements of the mother and child, but may be detectable in amniotic fluid.

In order to clarify why many amino acid concentrations in amniotic fluid correlated positively with maternal lipid intake and P-Ser correlated negatively, we analyzed the correlation between the intake of each type of lipid and the amino acid contents of the amniotic fluid. In particular, animal lipids, MUFAs, C16:0, C18:0, and C18:1 were found to have similar correlations to overall lipid intake. About 91% of the intake of MUFAs was C18:1, indicating that this correlation of MUFAs reflects the intake of C18:1. C16:0 is found in palm oil, shortening, butter, and other foods. It has been reported that excessive intake of C16:0 promotes fat accumulation and has a negative effect on the development and progression of type 2 diabetes [38]. C18:0 is found in animal fats and oils such as beef and pork. C18:1 is found in vegetable oils, such as olive oil and rape seed oil, but is also high in animal fats, such as beef. Among these FAs, C16:0 has been reported to promote serine synthesis through the expression of ATF4 in human cells [39,40]. Therefore, we believe that the negative correlation between C16:0 intake and P-Ser and the positive correlation between C16:0 intake and Ser (synthesized from P-Ser) is a result of the induction of serine synthesis in response to the intake of C16:0. In addition, studies using animal models have reported that a high-fat diet induces the expression of LAT1 and SNAT2 and activates the system A and system L transporters [41,42,43]. In the present study, all eight amino acids transported by system L (His/Met/Leu/Ile/Val/Phe/Tyr/Trp) were among the 17 amino acids that showed a significant (*p* > 0.05) correlation with lipid intake [44]. Amino acids transported by system A (Gly/Pro/Ala/Ser/Cysteine/Gln/Asn/Met/His) were also correlated, except for Gly, Pro, and cysteine [44]. Pro is an amino acid that was not detected during the analysis. Gly is rarely transported in the human placenta and is reported to be synthesized from serine in the fetus [45]. Cysteine was analyzed as cystine in this study, but no significant correlation was obtained. This may be due to the fact that cystine and cysteine are unstable in the acid used for pretreatment during analysis; thus, the values we obtained may be smaller than the actual values. The present results suggest that the intake of lipids, especially palmitic acid (C16:0), has an effect on amino acid concentrations in the amniotic fluid. Our results are consistent with previous reports indicating that high-fat diets in animal models activate the serine synthesis pathway and system A and L transporters.

We also analyzed the foods that correlated with the amino acid contents of the amniotic fluid and showed the top three correlation coefficients in two example cases (Figure 2). Egg intake was positively correlated with Asp, Phe, and Tyr levels. This suggests that these three amino acids, found to be correlated with lipid intake, may be a result of inclusion of eggs in the maternal diet. In addition, Trp, Leu, and Ser showed negative correlations with tsukemono intake. Tsukemono is a typical Japanese food with a low fat content and, as such, the negative correlation of the lipid intake dependent amino acids with tsukemono consumption was expected.

For the sample size of this study, if we calculate the recovery rate as 94% (calculated from the 36 participants initially recruited and the two individuals who did not complete the survey), the confidence level as 95% (λ = 1.96), and the sample error as 5–10 percentage points, the sample size ranged from 22 to 87 people. For conducting the analysis shown in Figure 2, the total of 34 subjects in this study was too small for many foods to be included. Although we were able to analyze foods that were generally eaten widely and daily, such as eggs, we were not able to analyze foods that were disliked by more than a certain number of the participants (such as bluefish) or seasonal foods. This small sample size was one of the limitations of this study.

## 8. Conclusions

The limitations of this study include the small number of subjects, the inclusion of the effects of anesthesia in the amniotic fluid and cord blood results due to the cesarean deliveries, maternal blood collection taking place at a different time from the amniotic fluid and cord blood collection, the high BMI of the subjects, and suspected underreporting due to low energy intake results from dietary survey. However, despite these limitations, the mother’s diet during pregnancy, especially fat intake, was positively correlated with many amino acid concentrations in the amniotic fluid and negatively correlated with P-Ser.

We believe that, by increasing the number of subjects and accumulating the results of dietary surveys and analysis of amniotic fluid components, it will be possible to clarify the effects of the habitual diet consumed by pregnant mothers on the nutrient contents (fatty acids, vitamins, minerals, etc.) and the associated metabolites in the amniotic fluid. In addition, clarifying the effects of maternal BMI and diseases such as gestational diabetes can also be addressed in future research. Furthermore, the impact of the amniotic fluid composition on the postnatal growth of the child will also be an issue for future studies. We believe that this information will be useful in providing nutritional guidance to pregnant women.

## Figures and Tables

**Figure 1 nutrients-13-02136-f001:**
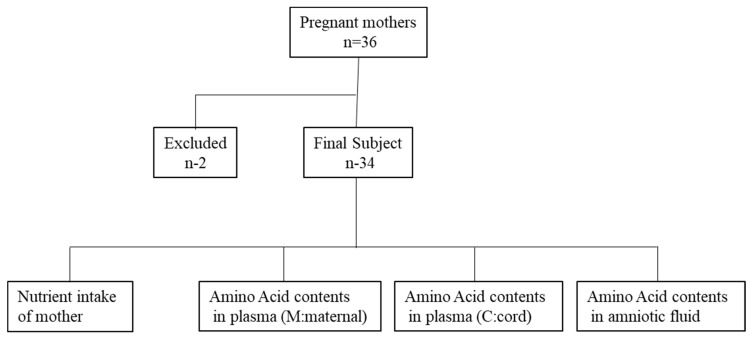
Flow chart of the study.

**Figure 2 nutrients-13-02136-f002:**
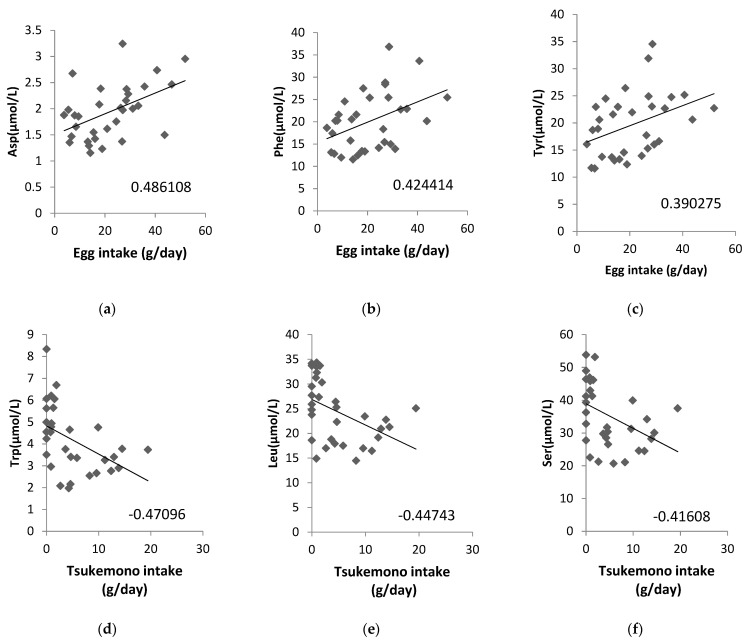
Correlations between specific maternal food intake and amino acid contents in the amniotic fluid. (**a**) Egg intake vs. Asp concentration, (**b**) egg intake vs. Phe concentration, (**c**) egg intake vs. Tyr concentration, (**d**) tsukemono intake vs. Trp concentration, (**e**) tsukemono intake vs. Leu concentration, and (**f**) tsukemono intake vs. Ser concentration. The correlation coefficients are indicated on each panel.

**Table 1 nutrients-13-02136-t001:** The average amino acid contents of each type of sample.

Amino Acids	Plasma (M)	Plasma (C)	Amniotic Fluid
	(μmol/L)	(μmol/L)	(μmol/L)
Essential	86.5 ± 15.6 ^a^	129.5 ± 22.7 ^b^	24.4 ± 6.3 ^c^
Leu	49.5 ± 8.7 ^a^	72.8 ± 10.5 ^b^	10.7 ± 3.3 ^c^
Ile	153.7 ± 25.2 ^a^	235.8 ± 32.0 ^b^	49.9 ± 16.2 ^c^
Val	163.9 ± 41.2 ^a^	270.3 ± 57.7 ^b^	98.7 ± 40.2 ^c^
Thr	39.5 ± 4.8 ^a^	72.2 ± 6.9 ^b^	4.2 ± 1.5 ^c^
Trp	51.5 ± 5.8 ^a^	70.3 ± 8.0 ^b^	20.1 ± 6.5 ^c^
Phe	24.8 ± 3.5 ^a^	28.2 ± 3.2 ^b^	8.3 ± 2.6 ^c^
Met	179.7 ± 34.2 ^a^	350.8 ± 48.3 ^b^	104.3 ± 32.1 ^c^
Lys	77.8 ± 13.5 ^a^	99.3 ± 8.9 ^b^	35.8 ± 8.9 ^c^
His	39.7 ± 7.9 ^a^	43.3 ± 5.8 ^a^	16.7 ± 4.7 ^b^
Non-essential	3.4 ± 0.6 ^a^	8.0 ± 7.1 ^b^	1.9 ± 0.5 ^c^
Asn	435.9 ± 74.4 ^a^	321.8 ± 61.9 ^b^	145.6 ± 49.2 ^c^
Asp	34.4 ± 9.1 ^a^	63.6 ± 12.4 ^b^	13.3 ± 4.5 ^c^
Ala	152.5 ± 34.9 ^a^	244.6 ± 43.8 ^b^	141.2 ± 46.6 ^a^
Arg	403.8 ± 66.0 ^a^	521.3 ± 74.8 ^b^	207.9 ± 73.0 ^c^
Gly	50.3 ± 13.4 ^a^	44.1 ± 19.3 ^a^	15.3 ± 6.1 ^b^
Gln	92.9 ± 18.5 ^a^	157.9 ± 15.5 ^b^	35.4 ± 9.8 ^c^
Glu	46.6 ± 7.7 ^a^	67.5 ± 8.9 ^b^	19.6 ± 5.8 ^c^
Ser	32.9 ± 6.3 ^a^	38.8 ± 6.6 ^b^	29.0 ± 7.3 ^c^
Tyr	1.5 ± 0.6 ^a^	2.2 ± 0.9 ^b^	6.8 ± 1.2 ^c^
Other	34.2 ± 5.1 ^a^	110.3 ± 29.4 ^b^	109.5 ± 29.8 ^b^
Cystine, μmol/L	1.4 ± 0.4 ^a^	7.6 ± 5.7 ^b^	6.3 ± 2.4 ^b^
P-Ser, μmol/L	933.5 ± 767.4 ^a^	1680.1 ± 1296.1 ^b^	2873.0 ± 2471.9 ^c^
Tau, μmol/L	1.7 ± 0.8 ^a^	2.0 ± 0.8 ^a^	2.0 ± 0.8 ^a^
PEA, μmol/L	921.7 ± 309.3 ^a^	1748.5 ± 383.2 ^b^	1501.2 ± 498.8 ^b^
Urea, μmol/L	13.6 ± 2.4 ^a^	15.9 ± 2.8 ^b^	5.6 ± 1.6 ^c^
Sar, μmol/L	23.2 ± 5.5 ^a^	22. 5 ± 4.8 ^a^	5.5 ± 1.6 ^b^
a-AAA, nmol/L	1.5 ± 0.4 ^a^	2.3 ± 0.5 ^b^	1.4 ± 0.4 ^a^
Cit, μmol/L	1.8 ± 0.8 ^a^	3.7 ± 1.4 ^b^	3.1 ± 1.1 ^b^
a-ABA, μmol/L	2.6 ± 1.2 ^a^	3.2 ± 2.2 ^ab^	4.7 ± 2.8 ^b^
Cystathio, μmol/L	607.9 ± 80.1 ^a^	814.8 ± 505.3 ^a^	798.3 ± 211.3 ^a^
b-Ala, μmol/L	4.6 ± 1.2 ^a^	27.3 ± 5.5 ^b^	47.1 ± 11.2 ^c^
b-AiBA, μmol/L	61.8 ± 15.5 ^a^	92.7 ± 25.9 ^b^	150.8 ± 55.1 ^c^
g-ABA, nmol/L	398.3 ± 298.1 ^a^	1624.2 ± 488.0 ^b^	1388.1 ± 415.6 ^b^
EOHNH_2_, μmol/L	36.6 ± 9.2 ^a^	81.4 ± 11.6 ^b^	13.1 ± 3.1 ^c^
NH_3_, μmol/L	ND	2.9 ± 1.7 ^a^	3.1 ± 1.6 ^a^
Hylys, nmol/L	1.9 ± 0.6 ^a^	2.4 ± 0.8 ^b^	2.0 ± 0.8 ^ab^
Orn, μmol/L	ND	1.7 ± 0.5	ND
1Mehis, μmol/L			
3Mehis, μmol/L			
Car, μmol/L			

Values are presented as mean ± SD. The table is divided into essential amino acids, non-essential amino acids, and others. Abbreviations used: C, cord; M, maternal; ND, not detected; Hyp, hydroxyproline; P-Ser, o-phosphoserine; Tau, taurine; PEA, o-phosphoethanolamine; a-AAA, α-aminoadipic acid; Cit, citrulline; a-ABA, α-aminobutyric acid; cystathio, cystathionine; b-Ala, β-alanine; b-AiBA, β-amino-iso-butyric acid; g-ABA, γ-aminobutyric acid; EOHNH_2_, ethanolamine; NH_3_, ammonia; Hyl, hydroxylysine; Orn, ornithine; His(1-Me), 1-methylhistidine; His(3-Me), 3-methylhistidine. Means followed by different letters are significantly different (*p* < 0.05) according to Tukey’s honestly significant difference (HSD) test and the Steel-Dwass test.

**Table 2 nutrients-13-02136-t002:** Spearman rank correlation coefficients of the amino acid contents of maternal plasma, cord blood plasma, and amniotic fluid samples with the maternal dietary intake of proteins, fats, and carbohydrates.

	Plasma (M)			Plasma (C)			Amniotic fluid				
	P (E%)	F (E%)	C (E%)	P (E%)	F (E%)	C (E%)	P (E%)	F (E%)	C (E%)		: 1
P-Ser	0.1272	0.2419	−0.2606	0.1976	0.1910	−0.3134	−0.0660	−0.3674 *	0.3036		: 0.75~1.00
Tau	0.0214	0.3545	−0.2952	0.0539	−0.1476	0.1081	0.0481	0.1918	−0.1392		: 0.50~0.75
PEA	0.1204	−0.2737	0.1998	−0.0633	0.0181	0.0315	0.0167	−0.1870	0.1763		: 0.25~0.50
Urea	0.2584	0.1826	−0.2703	0.3004	0.0941	−0.2757	0.2422	0.0984	−0.2285		: 0.00~0.25
Asp	−0.0101	−0.1141	0.1298	−0.0389	−0.0265	0.0425	0.1824	0.4075 *	−0.4709 **		: −1.00~−0.75
Thr	0.0784	0.0640	−0.0536	0.2673	0.2984	−0.3149	0.3398	0.3281	−0.3502 *		: −0.75~−0.50
Ser	−0.0461	−0.0978	0.1177	0.0781	0.3028	−0.2258	0.1105	0.3672 *	−0.3049		: −0.50~−0.25
Asn	0.0454	−0.0371	0.0546	0.1277	0.2450	−0.2918	0.1639	0.4763 **	−0.3985 *		: −0.25~0.00
Glu	0.1294	−0.3172	0.1942	−0.0883	−0.0963	0.1186	0.1165	0.3936 *	−0.3669 *		
Gln	−0.2495	0.0338	0.1013	−0.0020	−0.0017	0.0625	0.1489	0.4531 **	−0.3669 *		
Sar	−0.3077	0.0490	−0.0350	0.3578	0.1368	−0.2833	0.2675	0.3130	−0.3597		
a-AAA	−0.0299	−0.3419	0.3120	0.1961	0.2060	−0.1492	0.2838	0.0585	−0.0531		
Gly	−0.2460	−0.0485	0.2099	−0.0633	0.0715	−0.0300	0.2431	0.1612	−0.1805		
Ala	0.0289	0.3629 *	−0.3183	0.0020	0.2198	−0.2113	0.1284	0.3420	−0.3286		
Cit	0.0862	0.1437	−0.0920	0.0642	0.2594	−0.2310	0.1344	0.3654 *	−0.3437		
a-ABA	0.2919	−0.0101	−0.1094	0.2860	0.0099	−0.1680	0.3963 *	0.2515	−0.3421		
Val	0.1212	−0.1579	0.0793	0.1317	0.1020	−0.2063	0.1827	0.3304	−0.3137		
Cystine	0.0686	0.1227	−0.0613	0.0368	0.1967	−0.1478	0.2819	0.1343	−0.1670		
Met	−0.1197	−0.0481	0.1170	−0.1302	0.1947	−0.0660	0.1555	0.4891 **	−0.3993 *		
Cystathio	−0.0436	0.1100	−0.0764	−0.1597	0.0420	0.0353	0.4275 *	0.3545 *	−0.4392 *		
Ile	−0.0684	0.0660	−0.0042	−0.0055	−0.0486	−0.0175	0.1006	0.3071	−0.2891		
Leu	−0.0403	−0.0847	0.0938	0.1877	−0.0737	−0.0313	0.1210	0.2874	−0.2632		
Tyr	−0.0665	−0.1047	0.1120	0.0458	0.2324	−0.2618	0.1507	0.4565 **	−0.3978 *		
Phe	−0.0334	0.0264	0.0237	0.2712	0.2829	−0.4035 *	0.1776	0.5083 **	−0.4482 *		
b-Ala	−0.0526	0.0040	−0.0028	−0.0842	−0.0709	0.0374	0.2700	0.0066	−0.1039		
b-AiBA	−0.1434	−0.2377	0.3146	−0.0616	−0.2650	0.3051	0.1202	−0.1749	0.1064		
Trp	−0.2143	−0.0521	0.1835	0.2320	−0.2363	0.0852	0.0761	0.2517	−0.1802		
EOHNH_2_	−0.2166	0.2290	−0.0775	−0.2601	0.0309	0.0781	−0.0319	0.1539	−0.0732		
NH_3_	−0.0394	0.2350	−0.1371	0.1622	−0.0145	−0.0692	−0.0354	−0.0251	0.0423		
Orn	0.2112	−0.1840	0.0275	0.1128	0.0128	−0.0522	0.0983	0.1789	−0.1324		
Lys	0.0642	−0.1629	0.1146	0.1531	0.1618	−0.1581	0.2902	0.2276	−0.2119		
His	−0.0825	−0.0461	0.1213	−0.1601	0.1637	−0.0597	0.2307	0.4235 *	−0.3406		
His(3-Me)	0.3588 *	−0.1492	0.0590	0.3474 *	−0.0926	0.0411	0.2639	0.1081	−0.0819		
Arg	−0.2046	−0.0722	0.1394	0.1020	−0.1165	0.1690	0.0369	0.2958	−0.1978		

Abbreviations used: plasma (M), maternal plasma; plasma (C), cord plasma; P, protein; F, fat; C, carbohydrate; E%, percentage of daily energy intake; Hyp, hydroxyproline; P-Ser, o-phosphoserine; Tau, taurine; PEA, o-phosphoethanolamine; a-AAA, α-aminoadipic acid; Cit, citrulline; a-ABA, α-aminobutyric acid; cystathio, cystathionine; b-Ala, β-alanine; b-AiBA, β-amino-iso-butyric acid; EOHNH_2_, ethanolamine; NH_3_, ammonia; Orn, ornithine; His(3-Me), 3-methylhistidine. The numbers in the table show the correlation coefficients, with blue highlighting indicating positive correlations and pink highlighting indicating negative correlations. The correlation coefficients are divided into nine color-coded bins (from ±1 to 0). The darker the color, the higher the correlation. Significance level: * *p* < 0.05, ** *p* < 0.01.

**Table 3 nutrients-13-02136-t003:** Spearman rank correlation coefficients between lipid species in the maternal diet and amino acid concentrations in the amniotic fluid.

a								
	**Fat**	**Animal Fat**	**Vegetable Fat**	**Cholesterol**	**SFAs**	**MUFAs**	**PUFAs**	
Intake (g/day)	30.5	14.1	16.4	0.205	8.44	10.69	7.44	
Amino acids	correlation coefficient							
P-Ser	−0.3068	−0.2691	−0.2339	−0.1521	−0.3878 *	−0.3358	−0.1547	
Tau	0.1337	0.1624	0.0943	0.1719	0.2196	0.0976	0.0356	
PEA	−0.1347	−0.0058	−0.1042	−0.0244	−0.0569	−0.0805	−0.1817	
Urea	0.2517	0.1731	0.1173	0.2493	0.1862	0.2995	0.0448	
Asp	0.5587 **	0.3710 *	0.4216 *	0.5018 **	0.2991	0.5418 **	0.5198 **	
Thr	0.4017 *	0.1875	0.2002	0.0388	0.2574	0.3686 *	0.2299	
Ser	0.4842 **	0.3112	0.2848	0.1408	0.3215	0.4941 **	0.3229	
Asn	0.5729 **	0.4542 **	0.2790	0.3556 *	0.4732 **	0.5348 **	0.2980	
Glu	0.4820 **	0.4198 *	0.2677	0.3676 *	0.3399	0.4713 **	0.3559 *	
Gln	0.4923 **	0.4475 **	0.2062	0.1330	0.4756 **	0.3920 *	0.2289	
Sar	0.1494	0.3078	−0.0364	0.1091	0.3325	0.0286	0.1052	
a-AAA	0.0585	0.1523	−0.1838	0.2938	0.1785	0.0000	−0.2069	
Gly	0.2342	0.1484	0.0934	0.1780	0.1117	0.1994	0.1462	
Ala	0.4806 **	0.4439 **	0.2266	0.2473	0.3329	0.4920 **	0.3476 *	
Cit	0.4161 *	0.3207	0.1848	0.0652	0.3288	0.3794 *	0.2709	
a-ABA	0.3349	0.3269	0.1016	0.3419	0.2637	0.3697 *	0.1213	
Val	0.4348 *	0.2951	0.2390	0.1481	0.3232	0.4632 **	0.2406	
Cystine	0.1716	0.0445	−0.0035	0.0472	0.0891	0.1267	0.0830	
Met	0.6239 **	0.5022 **	0.3021	0.2621	0.4670 **	0.6034 **	0.3765 *	
Cystathio	0.3970 *	0.2031	0.2097	0.2060	0.2782	0.3226	0.2951	
Ile	0.4223 *	0.3438	0.1921	0.0960	0.3479	0.4520 **	0.1771	
Leu	0.3688 *	0.2815	0.1287	0.1345	0.2808	0.4128 *	0.1587	
Tyr	0.6301 **	0.5227 **	0.2980	0.3944 *	0.4432 *	0.6613 **	0.3563 *	
Phe	0.6444 **	0.5297 **	0.3369	0.3526 *	0.4754 **	0.6580 **	0.3911 *	
b-Ala	0.0005	0.0541	−0.1771	−0.1354	0.1387	−0.0043	−0.3082	
b-AiBA	−0.2990	−0.3586	−0.1143	−0.2133	−0.2394	−0.2650	−0.0946	
Trp	0.4230 *	0.4032 *	0.0480	0.1349	0.3424	0.4201 *	0.1056	
EOHNH_2_	0.1285	0.2788	−0.0909	0.2125	0.1740	0.0665	−0.0680	
NH_3_	0.0946	0.1331	0.0521	−0.0011	0.0718	0.1639	−0.0136	
Orn	0.2108	0.0433	0.1136	0.0590	0.0147	0.2023	0.2078	
Lys	0.2667	0.0572	0.1654	0.0588	0.1491	0.2517	0.1618	
His	0.5164 **	0.3676 *	0.3105	0.2550	0.3937 *	0.4853 **	0.3646 *	
His(3-Me)	0.1724	0.0478	0.1257	0.0416	−0.0354	0.1195	0.2707	
Arg	0.2625	0.3504	0.0343	0.2339	0.3423	0.2327	0.0585	
b								
	**n-3 FA**	**n-6 FA**	**C16:0**	**C17:0**	**C18:0**	**C18:1**	**C18:2 (n-6)**	**C20:3 (n-6)**
Intake (g/day)	1.36	6.07	4.92	0.072	1.79	9.76	5.90	0.017
Amino acids	correlation coefficient							
P-Ser	0.0084	−0.1785	−0.3622 *	−0.2738	−0.3486	−0.3548 *	−0.1734	−0.3013
Tau	−0.0374	0.0490	0.1820	0.3018	0.2483	0.1169	0.0399	0.1603
PEA	−0.2299	−0.1071	−0.0496	0.1063	0.0151	−0.0964	−0.1044	−0.0248
Urea	0.1668	0.0087	0.2557	0.3633 *	0.3185	0.2620	0.0120	0.2774
Asp	0.4186 *	0.5026 **	0.4571 **	0.2797	0.4351 *	0.5400 **	0.4897 **	0.5847 **
Thr	0.3252	0.2032	0.3061	0.2179	0.3178	0.3222	0.2049	0.2958
Ser	0.3284	0.3127	0.3827 *	0.3420	0.4102 *	0.4527 **	0.3083	0.3952 *
Asn	0.3845 *	0.2529	0.5007 **	0.4542 **	0.5297 **	0.5095 **	0.2390	0.5095 **
Glu	0.3168	0.3406	0.4188 *	0.3332	0.4268 *	0.4552 **	0.3255	0.5338 **
Gln	0.2577	0.2226	0.4586 **	0.4011 *	0.4739 **	0.3723 *	0.2199	0.4111 *
Sar	0.1325	0.0649	0.2169	0.1844	0.2273	0.0182	0.0260	0.1961
a-AAA	0.0538	−0.2615	0.1669	0.1485	0.1446	−0.0285	−0.2962	0.1662
Gly	0.2526	0.1261	0.1215	0.1435	0.1407	0.1581	0.1147	0.1896
Ala	0.3128	0.3396	0.3757 *	0.3817 *	0.3967 *	0.4576 **	0.3232	0.4539 **
Cit	0.4256 *	0.2262	0.3424	0.3589 *	0.3314	0.3380	0.2254	0.3270
a-ABA	0.2313	0.0959	0.3600 *	0.4171 *	0.4215 *	0.3235	0.0976	0.4906 **
Val	0.2500	0.2276	0.3580 *	0.3382	0.3914 *	0.4211 *	0.2320	0.3747 *
Cystine	0.2299	0.0634	0.0955	0.0738	0.0765	0.0967	0.0503	0.1031
Met	0.4417 *	0.3475	0.5198 **	0.5055 **	0.5411 **	0.5649 **	0.3332	0.5425 **
Cystathio	0.3013	0.3024	0.3207	0.2577	0.3054	0.3087	0.2724	0.3043
Ile	0.2606	0.1620	0.3889 *	0.4065 *	0.3860 *	0.4058 *	0.1584	0.3823 *
Leu	0.2416	0.1309	0.3266	0.3457	0.3548 *	0.3603 *	0.1276	0.3486
Tyr	0.3548 *	0.3402	0.5407 **	0.4967 **	0.5839 **	0.6268 **	0.3292	0.6309 **
Phe	0.3578 *	0.3831 *	0.5491 **	0.5440 **	0.5850 **	0.6213 **	0.3662 *	0.6063 **
b-Ala	−0.1323	−0.3134	0.1487	0.1262	0.1083	−0.0382	−0.3140	0.1183
b-AiBA	−0.0946	−0.0961	−0.3232	−0.2202	−0.2911	−0.2552	−0.1025	−0.3527
Trp	0.1976	0.0883	0.3930 *	0.3886 *	0.4227 *	0.3658 *	0.0777	0.4714 **
EOHNH_2_	0.0796	−0.1144	0.1725	0.2193	0.2327	0.0591	−0.1276	0.2043
NH_3_	−0.1309	0.0718	0.1573	0.1107	0.1939	0.1595	0.0975	0.1994
Orn	0.2940	0.2038	0.0674	0.0213	0.0660	0.1957	0.2108	0.1056
Lys	0.1965	0.1601	0.1725	0.1019	0.1698	0.2430	0.1604	0.1324
His	0.2888	0.3633 *	0.3907 *	0.3737 *	0.4609 **	0.4652 **	0.3523 *	0.3830 *
His(3-Me)	0.3406	0.2400	−0.0171	−0.0710	0.0056	0.1141	0.2392	0.0127
Arg	0.1710	0.0641	0.2980	0.3972 *	0.3101	0.2238	0.0452	0.2819

Abbreviations used in (**a**): FAs, fatty acids; SFAs, saturated fatty acids; MUFAs, monounsaturated fatty acids; PUFAs, polyunsaturated fatty acids. Abbreviations used in (**b**): n-3 FAs, n-3 fatty acids; n-6 FAs, n-6 fatty acids; C16:0, palmitic acid; C17:0, heptadecanoic acid; C18:0, stearic acid; C18:1, oleic acid; C18:2(n-6), linoleic acid; C20:3(n-6), eicosatrienoic acid; Hyp, hydroxyproline; P-Ser, o-phosphoserine; Tau, taurine; PEA, o-phosphoethanolamine; a-AAA, α-aminoadipic acid; Cit, citrulline; a-ABA, α-aminobutyric acid; cystathio, cystathionine; b-Ala, β-alanine; b-AiBA, β-amino-iso-butyric acid; His(3-Me), 3-methylhistidine. Intake represents the mean intake (g/day) across subjects. Significance level: * *p* < 0.05, ** *p* < 0.01.

**Table 4 nutrients-13-02136-t004:** Spearman rank correlation coefficients between dietary protein, meat, and egg intake and ammonia concentration in maternal plasma, cord blood plasma, and amniotic fluid samples.

NH_3_	Plasma (M)	Plasma (C)	Amniotic Fluid
P (%E)	−0.0394	0.1622	−0.0354
Chicken meat	0.0318	−0.0016	0.4817 **
Pork & beef meat	0.0160	0.0349	0.3959 *
Egg	0.1357	−0.3099	0.1411

Abbreviations used: plasma (M), maternal plasma; plasma (C), cord blood plasma; P (%E), protein as the percent of daily energy intake. Significance level: * *p* < 0.05, ** *p* < 0.01.

**Table 5 nutrients-13-02136-t005:** Baseline characteristics of mothers and infants (n = 34).

Mother	
Age, years	32.4 ± 4.6
Length, cm	157.1 ± 6.4
Bodyweight, kg	63.0 ± 12.2
BMI	25.5 ± 4.4
Parity	2.2 ± 0.7
Promipara, n (%)	3 (8.8)
Multipara, n (%)	31 (91)
BloodGlucose (mg/dL)	104.9 ± 21.4
BloodKetone (mmol/L)	0.1 ± 0.1
**Infants**	
Gestationalage (weeks)	38.1 ± 0.6
Birthweight (g)	2967 ± 275
GenderMale, n (%)	20 (58.8)
Female, n (%)	14 (41.2)
CordbloodGlucose (mg/dL)	63.6 ± 11.0
CordbloodKetone (mmol/L)	0.6 ± 0.2

Values are presented as mean ± SD.

**Table 6 nutrients-13-02136-t006:** Daily nutrient intakes reported on the BDHQs.

Nutrient	Mean±SD
Energy, kcal/day	1565 ± 384
Protein, %E	15.1 ± 2.1
Fat, %E	27.4 ± 4.5
Carbohydrate, %E	57.4 ± 5.6
Fiber, g/1000 kcal	6.9 ± 1.5
NaCl, g/1000 kcal	5.8 ± 0.9
Vitamin	428 ± 191
V.A, μg/1000 kcal	6.2 ± 3.3
V.D, μg/1000 kcal	4.0 ± 0.7
V.E, mg/1000 kcal	0.44 ± 0.06
V.B_1_, mg/1000 kcal	0.69 ± 0.14
V.B_2_, mg/1000 kcal	8.0 ± 1.6
Niacin, mg/1000 kcal	0.65 ± 0.11
V.B_6_, mg/1000 kcal	4.0 ± 1.3
V.B_12_, μg/1000 kcal	170 ± 44
Folate, μg/1000 kcal	3.74 ± 0.66

Abbreviations used: BDHQ, brief-type self-administered diet history questionnaire; %E, percentage of daily energy intake; V, vitamin; PaA, pantothenic acid; NaCl, table salt; K, potassium; Ca, calcium; Mg, magnesium; P, phosphorus; Fe, iron; Zn, zinc; Cu, copper; Mn, manganese. Energy intake is shown as daily energy intake. Protein, fat, and carbohydrate values are expressed as %E. All other values are shown as intake per 1000 kcal of total energy intake (energy-adjusted nutrient intake). Values are presented as mean ± SD.

## Data Availability

Not applicable.

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
