# Peer review of "Amino Acid Composition of Amniotic Fluid during the Perinatal Period Reflects Mother’s Fat and Carbohydrate Intake"

_nutrients, 2021, doi:10.3390/nu13072136_

Round 1
Reviewer 1 Report
The work presented in the current study is novel and will provide valuable descriptive data for future studies to reference. I have a few comments below and some suggestions for improving clarity.
1. As noted, This is a descriptive study, that assessed diet retrospectively. As such I would not make any causal statements, particularly in the abstract. Although placental transport of amino acids is a potential mechanism for their findings, the authors did not assess placental amino acid transport systems and thus this should not be the conclusion presented of their study. I would suggest they edit the manuscript to ensure that it is clear when they are proposing a potential mechanism versus what their data and study design can actually state.
2. Are there non-maternal sources that could impact the amino acid levels in amniotic fluid, for instance fetal synthesis?
Editing to improve clarity
3. Being discussion with major finding
4. Move paragraph 1 line 251 either to the results section or to a “strengths and limitations” section as this is mainly relevant to generalizability of the study sample and findings.
Sentence structure in the conclusion needs some work.
5. Line 337:“Our results further suggest that among lipids, palmitic acid intake in particular has an effect…”
Please clarify what effect palmitic acid has.
6. Line 338: “that lipid intake activates the serine synthesis pathway and system A and 338 system L transporters..”
There are animal models that would specifically test whether diet or protein consumption activate these systems. Please include some findings from animal models (sheep) to corroborate your suggestion that lipid intake activates system A and L.
7. Line: 339 “However, as it was suggested that the intake of pickled vegetables may increase serine in the amniotic fluid, …”
Why does this sentence start with ‘however’? Is this in reference to lipid intake?
8. Line 340: it is expected that by increasing the number of subjects and accumulating more data, future research will clarify which nutrients can be efficiently delivered to the fetus (i.e., with an increase in concentration in the amniotic fluid) through the consumption of specific foods.
This sentence should probably stand on its own and is a bit confusing as written. Please clarify.
Author Response
- As noted, This is a descriptive study, that assessed diet retrospectively. As such I would not make any causal statements, particularly in the abstract. Although placental transport of amino acids is a potential mechanism for their findings, the authors did not assess placental amino acid transport systems and thus this should not be the conclusion presented of their study. I would suggest they edit the manuscript to ensure that it is clear when they are proposing a potential mechanism versus what their data and study design can actually state.
Our goal was to determine whether the habitual diet of a free-living pregnant human woman would affect amniotic fluid composition, as has been shown in studies using animal models. Following your advice, we have removed the statement about amino acid transport from the Conclusion and have revised the Discussion. We rewrote the Conclusion and Discussion, as well as the Abstract.
- Are there non-maternal sources that could impact the amino acid levels in amniotic fluid, for instance fetal synthesis?
It has been reported that the human embryo (in organs such as the liver and small intestine, for example) expresses enzymes involved in amino acid and ammonia metabolism and can catabolize amino acids to produce energy. This, we believe, also affects the amino acid concentrations in the amniotic fluid. We have included a description of this in the Discussion (ref 33).
Editing to improve clarity
- Being discussion with major finding
We have re-written the Discussion and Conclusion, incorporating the advice from your first suggestion.
- Move paragraph 1 line 251 either to the results section or to a “strengths and limitations” section as this is mainly relevant to generalizability of the study sample and findings.
We have followed your advice and have moved the first sentence to the results.
Sentence structure in the conclusion needs some work.
- Line 337:“Our results further suggest that among lipids, palmitic acid intake in particular has an effect…
Please clarify what effect palmitic acid has.
In response to your comment, we have added to the Discussion and have cited references for the statement that excessive palmitic acid intake promotes fat accumulation and plays a detrimental role in the development and progression of type 2 diabetes.
- Line 338: “that lipid intake activates the serine synthesis pathway and system A and 338 system L transporters..”
There are animal models that would specifically test whether diet or protein consumption activate these systems. Please include some findings from animal models (sheep) to corroborate your suggestion that lipid intake activates system A and L.
We could not find any papers using sheep, but we added one review (ref 43).
- Line: 339 “However, as it was suggested that the intake of pickled vegetables may increase serine in the amniotic fluid, …”
Why does this sentence start with ‘however’? Is this in reference to lipid intake?
"However" has been removed and the sentence has been edited for clarity.
- Line 340: it is expected that by increasing the number of subjects and accumulating more data, future research will clarify which nutrients can be efficiently delivered to the fetus (i.e., with an increase in concentration in the amniotic fluid) through the consumption of specific foods.
This sentence should probably stand on its own and is a bit confusing as written. Please clarify.
We have re-written the conclusion in response to your suggestion.
Reviewer 2 Report
The authors assess plasma, cord blood and amniotic fluid amino acid concentrations in relation to habitual dietary intake during the final month of pregnancy. All participants delivered their infants by c-section. The manuscript presents a large set of data.
Methods should indicate if the p-values presented are false-discovery rate corrected. It should be clearly indicated if these p-values have not been corrected for multiple comparisons.
Table 4 should include a marker of statistical significance for the correlations presented. Perhaps an * or dagger, such as used in Tables 5 & 6.
The discussion would be strengthened by comparison of the amino acid levels in cord blood, amniotic fluid, and maternal plasma to that of the average adult Japanese woman with similar BMI or just general Japanese adult.
Author Response
Methods should indicate if the p-values presented are false-discovery rate corrected. It should be clearly indicated if these p-values have not been corrected for multiple comparisons.
FDR was not used in this analysis. We have added a column of statistics to show this.
Table 4 should include a marker of statistical significance for the correlations presented. Perhaps an * or dagger, such as used in Tables 5 & 6.
We have followed your advice and put asterisk (*) marks in Table 4 as well.
The discussion would be strengthened by comparison of the amino acid levels in cord blood, amniotic fluid, and maternal plasma to that of the average adult Japanese woman with similar BMI or just general Japanese adult.
We could not find any papers that examined the amino acid contents in the amniotic fluid or blood samples of Japanese people, so we could not include this comparison in our Discussion. We would like to continue further research to clarify the nutritional impact of BMI on the amino acid contents of amniotic fluid and have added this information to the future issues section.
Reviewer 3 Report
The manuscript is interesting and well written. The main finding of this study is that several amino acids in the amniotic fluid were positively correlated with lipid intake, but not with protein intake. The methodology is correct. The conclusions are supported by the results. The limits of the study are sufficiently commented on.
Please find the below comments and queries that should be incorporated and addressed before finalizing the decision on your manuscript.
-line 73: "Daily nutrient intake was explored using a brief, self-administered diet history questionnaire (BDHQ)": This tool should be better explained in the methods section ( who explained method to participants? dietitians? MD? Was supplement intake recorded? What about portion size? ); Discuss also limitations of this tool.
- Were mother body weight or BMI present in the medical records or self-declared? please clarify and eventually discuss among limitations. Are there other body composition measurments in the clinical records? Why not?
-line 251-257- The presentation of the results must be well separated from the discussion of the results
-Statistics section is poor . Expand this part, explaining all the statistical analisys used
-What about sample size calculation? Add this part.
Author Response
Please find the below comments and queries that should be incorporated and addressed before finalizing the decision on your manuscript.
-line 73: "Daily nutrient intake was explored using a brief, self-administered diet history questionnaire (BDHQ)": This tool should be better explained in the methods section ( who explained method to participants? dietitians? MD? Was supplement intake recorded? What about portion size? ); Discuss also limitations of this tool.
A dietitian was present during the meal survey, explaining how to fill out the form and what to look out for and answering questions during the process of filling out the form. After the participants filled out the form, we reviewed the written surveys to see if there were any omissions. The dietitian also asked the participants to answer the survey questions verbally to make sure there were no omissions. We have followed your advice and added these descriptions to the Methods section. We have also added a discussion of the limitations of dietary surveys in general. Accordingly, we have added "description of the food survey" to the Author Contributions.
- Were mother body weight or BMI present in the medical records or self-declared? please clarify and eventually discuss among limitations. Are there other body composition measurments in the clinical records? Why not?
We asked the mothers to self-report their weight and height at the time of the dietary survey and then calculated their BMI from this information. It may vary by country, but in Japan, weight after delivery is not recorded in the medical records. Other than the mother's height and weight (before delivery), no other data on body composition was available in the data we had access to. We added the fact that height and weight were self-reported as a limitation of the study.
-line 251-257- The presentation of the results must be well separated from the discussion of the results
We have moved the indicated lines from the Discussion to the Results.
-Statistics section is poor . Expand this part, explaining all the statistical analisys used
As you have pointed out that the statistics section was insufficient, we have expanded the information provided.
-What about sample size calculation? Add this part.
We did not describe the sample size calculation because this was not a widely conducted general survey. The survey was conducted by recruiting pregnant women who were planning to give birth by cesarean section at the cooperating hospital during the maximum dates of the study period. The maximum number of potential participants who could consent to participate was 36; these women were the subjects of this study. Of these 36, the collection rate was 94%, after excluding the two women who were not able to complete the dietary survey. Based on this, if we set the confidence level at 95% (λ = 1.96) and the sample error at 5 percentage points, the sample size is 87 people. If the sample error is calculated as 10 percentage points, the sample size is 22. We have added a statement about this sample size calculation into the Discussion section.
Round 2
Reviewer 3 Report
No further comments.